# Recent Advances in the Therapeutic Management of Advanced Systemic Mastocytosis

**DOI:** 10.3390/diagnostics14010080

**Published:** 2023-12-29

**Authors:** Scott Veitch, Deepti H. Radia

**Affiliations:** Department of Haematology, Guy’s and St Thomas’ NHS Foundation Trust, London SE1 7EH, UK

**Keywords:** advanced systemic mastocytosis, tyrosine kinase inhibitors, KIT inhibitors, midostaurin, avapritinib, bezuclastinib

## Abstract

Advanced systemic mastocytosis (AdvSM) is a rare haematological neoplasm characterised by the accumulation of neoplastic mast cells (MCs) in various organs, resulting in organ dysfunction and reduced life expectancy. The subtypes include aggressive SM (ASM), SM with an associated haematological neoplasm (SM-AHN) and mast cell leukaemia (MCL). The gain of function *KIT* D816V mutation is present in most cases. The availability of tyrosine kinase inhibitors (TKIs) has revolutionised the treatment landscape for patients with this life-limiting disease. Patients are now able to achieve molecular remission, improved quality of life and improved overall survival. This review focuses on the targeted therapies currently available in clinical practice and within the clinical trial setting for AdvSM. This review also highlights possible future therapeutic targets and discusses therapeutic strategies for this multimutated and clinically heterogeneous disease.

## 1. Introduction

Systemic mastocytosis (SM) is a rare, clonal haematological neoplasm resulting in the expansion and accumulation of neoplastic mast cells (MCs) in multiple organs, including the skin, spleen, liver, bone marrow and gastrointestinal tract. Central to the pathogenesis of SM is the somatic *KIT* D816V mutation, which results in the constitutively activated ligand-independent transmembrane receptor kinase c-KIT and is present in 95% of cases [1]. SM is one of three subtypes of mastocytosis recognised in the fifth edition of the World Health Organization classification of myeloid neoplasms [2], the other two being cutaneous mastocytosis (CM) and mast cell sarcoma (MCS). CM is characterised by mast cell infiltration limited to the skin, whereas SM is characterised by MC infiltration within one or more extracutaneous organs.

The sub-classification of SM is complex and can present diagnostic challenges for the clinician. Six SM subtypes exist: bone marrow mastocytosis (BMM), indolent SM (ISM), smouldering SM (SSM), aggressive SM (ASM), mast cell leukaemia (MCL) and SM with an associated haematological neoplasm (SM-AHN). The latter three (ASM, MCL and SM-AHN) fall under the umbrella of ‘advanced SM’ (AdvSM) and were associated with an overall poor prognosis, ranging from 2 to 4 years for SM-AHN and <6 months for MCL, in the pre-tyrosine-kinase-inhibitor (TKI) era [3,4,5]. The presence or absence of B-findings (indicating a high burden of disease), C-findings (indicating MC-induced end organ damage, requiring cytoreductive therapy) or an AHN largely help define the SM subtype. Bone marrow mastocytosis is now a separate SM subtype, according to the fifth edition of the WHO classification of myeloid neoplasms, and is characterised by the absence of skin lesions and B-findings, a serum tryptase concentration below 125 mcg/L and often shows a male predominance and a strong association with severe allergic reactions to Hymenoptera stings. Of note, BMM is not an entity recognised by the International Consensus Committee 2022 (ICC) [6]. Aggressive SM (ASM) is defined by the presence of one or more C-findings. To qualify for a diagnosis of SM-AHN, the patient must meet the diagnostic criteria for SM and the criteria for an associated haematological neoplasm, which are generally myeloid in origin. The most common AHN is chronic myelomonocytic leukaemia (CMML); others include myelodysplastic syndrome (MDS), myeloproliferative neoplasm (MPN), MDS/MPN overlap syndrome and acute myeloid leukaemia (AML). Mast cell leukaemia (MCL) is historically defined by ≥20% atypical immature MCs in the bone marrow aspirate, according to the WHO HAEM-5 classification. However, the non-uniform localization of mast cell infiltrates, patchy areas of fibrosis or suboptimal bone marrow aspirate techniques can lead to discrepancies between aspirate and core biopsy findings. Consequently, the International Consensus Committee (ICC) introduced a provision to allow the diagnosis of MCL from a core biopsy alone, provided that the core biopsy shows a dense, diffuse infiltration of atypical immature MCs [6]. Additional refinements to the diagnostic criteria have been made, largely based on morphology (Table 1). Immunohistochemistry demonstrating the presence of tryptase, CD117 and CD25 +/− CD2 positivity is used for the clonal identification of mast cell aggregates (Figure 1). Additionally, aberrant CD30 expression has been added as a minor diagnostic criterion. In addition to the detection of the *KIT* D816V mutation, the detection of any activating KIT mutation has been added as a minor criterion. In cases of known hereditary alpha-tryptasaemia, the tryptase concentration should be adjusted in line with the number of extra copies of the alpha tryptase gene.

### Treatment Principles

On confirmation of the diagnosis, an individualised and multi-disciplinary treatment approach in a specialised centre is recommended. All patients should be counselled on strategies to avoid known triggers of MC degranulation. The risks of anaphylaxis should be highlighted, and it is recommended that all patients carry an adrenaline autoinjector. There is a range of available anti-mediator therapies [7] (e.g., H1 and H2 antihistamines, proton pump inhibitors, leukotriene receptor inhibitors, cromolyn and corticosteroids) aimed at reducing the symptoms caused by MC mediator release, and this is well documented. Bone disease should be detected using dual-energy X-ray absorptiometry (DEXA) and treated according to severity (e.g., bisphosphonates for osteoporosis). Immune tolerance therapy is recommended for those with positive hymenoptera-specific IgE skin testing, and anti-IgE treatment, such as Omalizumab, has been shown to be of benefit in a small cohort of patients [8].

## 2. Advanced SM

### 2.1. Risk Stratification

AdvSM is a multimutated disease, with somatic mutations beyond *KIT* D816V identified in approximately 90% of patients. The most frequently affected genes include *TET2, SRSF2, ASXL1, RUNX1* and *CBL* [9]. Mutated genes in the S/A/R panel (*SRSF2, ASXL1* and *RUNX1*) are associated with inferior overall survival (OS) in SM patients and are considered high-risk mutations [10]. Moreover, the number of additional somatic mutations will affect the patient’s overall outcome. An analysis of 70 SM patients demonstrated a clear difference in 3-year OS between patients with no mutations (90% OS), patients with one mutation (73% OS) and patients with two or more mutations (42% OS) in the S/A/R gene panel [11].

Prognostic scoring systems integrating clinical and molecular characteristics for patients with SM have evolved (Table 2). These are not yet embedded into routine clinical practice but can guide clinicians in assessing the overall prognosis of individual patients. These include the International Prognostic Scoring System for Mastocytosis (IPSM) [12], the Mayo Alliance Prognostic System (MAPS) [13], the Mutation-Adjusted Risk Score (MARS) [14] and the Global Prognostic Score for Mastocytosis (GPSM) [15]. A multivariate analysis of the validated MARS scoring model for patients with AdvSM (*n* = 383) identified that inferior OS was associated with age >60 years, anaemia (Hb <100 g/L), thrombocytopenia (platelet count <100 × 10^9^/L) and the presence of one or more high-risk mutations (S/A/R panel). High-risk, intermediate-risk and low-risk groups were devised based on these parameters, with respective OS times of 1.9 years, 4.3 years and OS not reached. The GPSM scores for PFS and OS, derived from the analysis of 1275 SM patients, are based on independent prognostic biomarkers for PFS (platelet count <100 × 10^9^/L, serum β2 microglobulin ≥2.5 μg/L and baseline serum tryptase ≥125 μg/L) and OS (Hb <110 g/L, alkaline phosphatase ≥140 IU/L and S/A/R/*DNMT3A* mutations).

### 2.2. Therapeutic Management Strategies

The availability and clinical impact of tyrosine kinase inhibitors (TKIs) as targeted treatments have revolutionised the management of patients with AdvSM over the last decade. Prior to this, cytoreductive therapy was the mainstay of treatment to reduce the mast cell burden, symptom burden and degree of end organ damage where possible. This review focuses on the tyrosine kinase inhibitors that are currently available in clinical practice and within trials. Additionally, it offers a brief summary of cytoreductive treatments, which may be an alternative for patients who may not be able to access TKIs (Figure 2).

### 2.3. Cytoreductive Therapy

Interferon alpha (IFN-α) has shown mixed results in a small number of retrospective studies, as well as a lack of durable responses [16,17]. IFN-α with or without prednisolone was studied in 47 patients. The ORR was 53%, the median duration of response was one year and only 3% achieved CR [18]. The toxicities of IFN-α include thyroid dysfunction, neuropsychiatric symptoms, flu-like symptoms and myelosuppression, which often precludes meaningful treatment compliance. The role of IFN-α has been superseded with the availability of TKIs.

Cladribine (2-chlorodeoxyadenosine [2-CdA]) can be used for fast disease debulking prior to initiating targeted TKI treatment or if a patient does not meet the eligibility criteria for TKI use, e.g., significant cardiac disease or thrombocytopenia with platelet counts <50 × 10^9^/L. An ORR of 50% and a median duration of response of 2.47 years was reported (with a range from 0.5 to 8.6 years) in a cohort of 32 AdvSM patients treated with 0.14 mg/kg of intravenous or subcutaneous infusions of Cladribine over 5 days, repeated every 4–12 weeks [19]. The most common adverse events were lymphopenia (82%), neutropenia (47%) and opportunistic infections (13%). However, significant reductions in serum tryptase concentrations were not observed in the AdvSM cohort and data on bone marrow responses were limited.

### 2.4. Tyrosine Kinase Inhibitors

The aims of KIT inhibitors are as follows:Limit and/or reverse MC-related organopathy;Target both the SM and the AMH clone, in cases of multilineage involvement by *KIT* D816V.

#### 2.4.1. Imatinib

In vitro studies have demonstrated that imatinib inhibits wild-type KIT, as well as KIT mutations outside of exon 17, such as in the intracellular juxtamembrane region or transmembrane region (e.g., Phe522Cys, Val560Gly and Lys509Ile) [20,21,22]. However, it is not active against *KIT* D816V [23]. Imatinib is known to be efficacious in the treatment of other c-KIT mutated haematopoietic and non-haematopoietic neoplasms, such as gastrointestinal stromal tumours (GISTs) and chronic myeloid leukaemia (CML), as well as in patients with SM and coexisting chronic eosinophilic leukaemia (SM-CEL) associated with platelet-derived growth factor (*PDGFR*) α and β rearrangements [24]. In 2006 it was approved by the FDA for use in adults with ASM lacking the *KIT* D816V mutation or with unknown KIT mutational status. Its efficacy was observed in D816V-negative well-differentiated systemic mastocytosis (WDSM) patients treated with 300 or 400 mg daily for 12 months, demonstrating an ORR of 50% [25,26]. Its utility is limited to a very small cohort of SM cases lacking the *KIT* D816V mutation.

#### 2.4.2. Midostaurin

Midostaurin is a multikinase inhibitor with in vitro activity against a variety of targets, including Fms-related tyrosine kinase 3 (FLT3), kinase domain *KIT* D816V, *PDGRF*α, *PDGFR*β and *VEGF* mutations [27]. Midostaurin was approved in 2007 for treating patients with AdvSM following a global, single-arm phase 2 study that recruited 116 patients, of whom 89 were considered evaluable for response (16 ASM, 16 MCL and 57 SM-AHN) with at least one measurable C-finding [28,29]. The patients received 100 mg of midostaurin twice daily until disease progression or unacceptable toxicity. With a median follow-up of 26 months (with a range of 12–54 months), the ORR was 60% (45% with a major response and 15% with a partial response). Using modified Valent and Cheson criteria, the response rates reported were highest in ASM patients (75%), with 58% and 50% of those with SM-AHN and MCL responding, respectively. The patient’s past treatment history or *KIT* D816V mutation status did not affect the outcomes. The median duration of response was 24.1 months, and the median overall survival was 28.7 months. The median best percentage change in bone marrow MC burden was −59%, and the median change in serum tryptase was −58%. Of all the subjects, 77% demonstrated a reduction in spleen volume. An objective improvement in disease-related symptoms and a reduction in the severity of cutaneous manifestations were noted. Mild gastrointestinal adverse events were reported in 82% of patients at all grades, with 6–8% reporting grade 3–4 symptoms. Nausea and vomiting can generally be prevented with the use of prophylactic anti-emetics prior to each dose. The most frequently observed grade 3 or 4 haematological toxicities were neutropenia (24%), anaemia (41%) and thrombocytopenia (29%).

#### 2.4.3. Avapritinib

Avapritinib is a type 1 multikinase inhibitor which was developed to selectively target *KIT* D816V and other KIT exon 17 mutations (e.g., homologous PDGFRA-mutant proteins). Avapritinib was investigated in patients with AdvSM first in the pivotal phase 1 EXPLORER study (NCT02561988) [30], followed by the international phase 2 registrational PATHFINDER study (NCT03580655) [31]. This led to its approval by the FDA in June 2021 and the European Medicines Agency in March 2022 for patients with AdvSM with a platelet count >50 × 10^9^/L. An interim analysis was performed on 32 response evaluable patients within the PATHFINDER study using the modified International Working Group-Myeloproliferative Neoplasms Research and Treatment (mIWG-MRT-ECNM) criteria. With a median follow-up of 10.4 months, the confirmed ORR (CR/CRh/PR/CI; primary endpoint) was 75% (*n* = 24), with 19% (*n* = 6) achieving CRh. The median time to response was 2 months. Responses were observed in all AdvSM subtypes and were not affected by the presence of S/A/R mutations. A reduction of ≥ 50% in serum tryptase from baseline was observed in 93% of patients, with similarly significant responses in bone marrow mast cell burden (88%) and *KIT*-D816V variant allele frequency (VAF) (60%). Of all the patients with SM-CMML, 80% exhibited a decrease of ≥50% in their absolute monocyte counts, and 88% of patients with eosinophilia had a decrease of ≥50% in the absolute eosinophil count. The most frequently reported AEs were peripheral and periorbital oedema (50% and 48%, respectively), primarily graded as 1–2 and managed with diuretics and dose reductions. Grade 3 neutropenia, thrombocytopenia and anaemia was seen in 24%, 16% and 16% of patients, respectively. The gastrointestinal AEs reported were predominantly grade 1/2, including diarrhoea (23%), nausea (18%) and vomiting (18%) [32,33].

#### 2.4.4. Bezuclastinib

Bezuclastinib is a novel, highly selective and potent *KIT* D816V inhibitor. Bezuclastinib has minimal brain penetration, favourable pharmacokinetic properties and has been designed to spare closely related kinases, e.g., PDGFRα, PDGFRβ, wild-type KIT, VEGFR2 (KDR) and CSF1R (FMS). Hence, it aims to minimise toxicities, such as bleeding, oedema and pleural effusions [34]. APEX is a phase 2 multicentre, two-part, open-label clinical study evaluating the safety and efficacy of bezuclastinib in patients with AdvSM [35]. Key entry criteria included a diagnosis of ASM, SM-AMH or MCL, the presence of one or more measurable C-findings per mIWG-MRT-ECMN and a platelet count of >50 × 10^9^/L. In part 1, the subjects were randomised to one of four doses. Part 2 is an expansion phase. Preliminary data have shown that it is generally well tolerated across all dose levels, presenting a favourable safety profile and displaying encouraging early signs of clinical activity. An interim analysis of 16 patients (three with ASM, twelve with SM-AHN and one with MCL) with a median age 69 years, demonstrated no related cognitive or bleeding AEs. The ORR (CR/CRh/PR/CI by mIWG-MRT-ECNM) was 89% in TKI therapy-naïve patients and 73% in all patients at a median follow-up of 27 weeks. A >50% reduction in serum tryptase, bone marrow MC burden and *KIT* D816V VAF was observed in 88%, 100% and 92% of the subjects, respectively. The most common non-haematological AEs were hair colour change (25%), taste disorders (25%) and peripheral oedema (19%). Haematological toxicities were predominantly low grade, reversible and did not require dose modification (neutropenia, 25%; thrombocytopenia, 19%; anaemia, 13%). Enrolment for the Part 2 expansion phase is ongoing.

An international, double-blind, placebo-controlled, phase 2 study (Summit; NCT05186753) is also underway, aiming to evaluate the safety, efficacy and biomarker correlates of bezuclastinib in patients with ISM and SSM [36].

#### 2.4.5. Elenestinib

Elenestinib (BLU-263; Blueprint Medicines) is a potent, selective, small molecule TKI with a high degree of selectivity for *KIT* D816V. A phase 1 study in normal healthy volunteers demonstrated its safety, with linear pharmacokinetics across all tested doses and a half-life that enables once-daily dosing. Another favourable property is its suggested limited central nervous system penetration (Kp,uu 0.06), which may reduce the risks of cognitive toxicities and intracranial bleeding. The double-blind placebo-controlled phase 2/3 HARBOR study (NCT04910685) is currently evaluating its efficacy in ISM patients [37]. There is significant clinical interest in evaluating BLU-263 in AdvSM patients, either as a standalone agent or in combination therapies.

### 2.5. Combination Therapy

SM-AHN displays significant genetic heterogeneity and an often unpredictable disease trajectory. Patients who harbour high-risk mutations may benefit from hypomethylating agents (HMAs) to treat the AHN component (most commonly CMML), but the decision regarding which component to treat is challenging and relies on clinical judgement to ascertain the primary driver of the patient’s symptoms and organopathy. Combining an HMA with a *KIT* D816V inhibitor in high-risk patients represents a potential therapeutic approach to improve responses by targeting both clones. No prior studies have evaluated this treatment approach. This led to the development of AZURE, a multicentre phase 1/2, open-label trial aimed at evaluating the safety and efficacy of the oral *KIT* D816V inhibitor BLU-263 in patients with AdvSM. BLU-263 is either given as monotherapy or in combination with azacitidine (75 mg/m^2^/day on days 1–7 of a 28-day cycle) [38]. Patients eligible for the combination therapy include those with CMML-2, MDS with excessive blasts-2, high- or very-high-risk myelodysplastic syndrome (MDS) as determined by the revised International Prognostic Scoring System for Myelodysplastic Syndromes (IPSS-R), complex karyotype/mutational profiles or accelerated phase myeloproliferative neoplasms. The study is currently recruiting internationally.

With this in mind, recent amendments to the APEX study in the part 2 expansion phase include the addition of an arm for high-risk AdvSM patients, aiming to recruit patients into a combination therapy option that includes Azacytidine.

### 2.6. Allogeneic Haematopoietic Stem Cell Transplantation

To date, the definitive role of allogeneic haematopoietic stem cell transplant (alloHSCT) has not been assessed by a large prospective trial. In a large retrospective study of 57 patients with AdvSM treated with alloHSCT (38 SM-AHN, 7 ASM and 12 MCL), CR was observed in 16% of patients. Furthermore, 70% of patients exhibited a reduction in BM MC burden, serum tryptase levels and organ involvement [39]. The overall survival (OS) was 57% for all the subjects at 3 years (SM-AHN, ASM and MCL: 74%, 43% and 17%, respectively). The median response duration time was 20 months. Progression free survival (PFS) was 57% at 1 year and 51% at 3 years. The risk factor most strongly associated with poor OS was MCL, with a median survival of 6 months. Myeloablative conditioning (MAC) was associated with better outcomes than reduced-intensity conditioning (RIC) in patients with AdvSM. The treatment-related mortality was 11% at 6 months and 20% at 12 months.

Several unanswered questions remain, such as determining the optimal debulking strategies pre-transplant, identifying the ideal conditioning regimens, understanding the role of MRD monitoring for detectable *KIT* D816V post-transplant and defining preferred candidates for alloHSCT. Furthermore, in the era of KIT inhibitors, questions persist regarding their role in pre-transplant debulking and if/when to reinitiate KIT inhibition post-transplant.

In summary, the available data suggest that alloHSCT is a potentially curative therapeutic option for AdvSM, but there is an unmet need for international collaborative prospective trials in this field to fully elucidate the true value of alloHSCT and to define its role in the era of KIT inhibitors.

## 3. Non-Advanced SM

Until recently, the mainstay of treatment for patients with a diagnosis of indolent systemic mastocytosis has been symptomatic, utilising combinations of anti-mediator therapies available for each patient. The overall prognosis and life expectancy of ISM patients are comparable to those of a healthy population, but this assessment does not consider that a significant proportion of patients experience poor quality of life due to the burden of their symptoms. A US-based survey of 56 SM patients focused on questions about their diagnosis, heath care use, quality of life and symptom burden [40]. A total of 93% experienced ≥10 SM-related symptoms, 54% reported reducing their work hours because of SM and 64% reported that they avoided leaving their home because of their diagnosis. The mean total symptom score for the Indolent Systemic Mastocytosis Symptom Assessment Form (ISM-SAF) was 52.7; a score of ≥28 is used to indicate moderate-to-severe symptoms. Over a 1-year recall period, 30% reported an episode of anaphylaxis requiring emergency care. A larger EU-based survey [41] is currently taking place, and recent interim data from 98 ISM patients demonstrates a substantial burden of disease and poor quality of life in this cohort.

Avapritinib is the first targeted therapy approved for ISM patients, representing a shift from largely supportive care involving anti-mediator therapy to disease-modifying therapy. Avapritinib is a selective *KIT* D816V inhibitor. It was approved by the FDA in May 2023 and the European Medicines Agency in November 2023 for the treatment of symptomatic ISM patients. The PIONEER trial [42,43] (Blueprint Medicines Corporation; NCT03731260) is a double-blind, placebo-controlled, phase 2 trial. It randomized patients with moderate-to-severe ISM (total symptom score [TSS] ≥ 28) in a 2:1 ratio to receive either 25 mg of avapritinib once daily (*n* = 141) or a placebo (*n* = 71), both with best supportive care (BSC). The primary endpoint was the mean change in TSS (range 0–110), based on the 14-day average of self-reported severity of 11 symptoms among the patients. Secondary endpoints included a ≥50% and ≥30% reduction in TSS; ≥50% reductions in serum tryptase, blood *KIT* D816V VAF and bone marrow mast cells; and quality of life (QoL) measures. The primary and key secondary endpoints were assessed from baseline to week 24. Avapritinib significantly improved the TSS (mean change −15.6 vs. −9.2; *p* = 0.003), and the likelihood of achieving a reduction in TSS of ≥50% (25% vs. 10%; *p* = 0.005) and ≥30% (45% vs. 30%; *p* = 0.009) compared to the placebo. A greater proportion of avapritinib-treated patients achieved reductions of ≥50% in serum tryptase, *KIT* D816V VAF and bone marrow mast cell burden (all *p* < 0.0001). The QoL scores showed up to 4.1-fold greater improvement with avapritinib compared to the placebo. The safety profiles were similar between the treatment groups, with few discontinuations due to adverse events. Avapritinib (25 mg once daily) appeared to be well tolerated by the majority of patients.

Further trials for symptomatic ISM patients with tyrosine kinase inhibitors are currently in progress and we await their results [36,37,44].

## 4. Recommended Follow-Up

It is recommended that all patients have a baseline bone marrow aspirate and trephine biopsy to confirm the diagnosis, especially of both components in the context of SM-AHN. Integrating morphology and immunohistochemistry with cytogenetic and molecular profiles and aligning these with clinicopathological findings (B- and C-findings) enables correct classification of the patient and informs treatment options. Follow-up studies are individually driven. Clinical practice at our centre of excellence is to repeat the bone marrow assessment and imaging studies 3–6 months after a therapeutic intervention to assess changes in the degree of organomegaly if present, histological bone marrow MC disease burden and molecular monitoring (changes in VAF). Increases in the VAF of pre-existing somatic mutations or the development of emergent mutations informs the potential need to change the therapeutic agent. Therefore, incorporating serial Next Generation Sequencing (NGS) into clinical practice is recommended. The parallel monitoring of c-KIT VAF should be carried out using highly sensitive PCR methods.

The treatment response criteria for AdvSM have been updated [45,46]. The modified International Working Group-Myeloproliferative Neoplasms Research and Treatment (IWG-MRT) and the European Competence Network on Mastocytosis (ECNM) consensus response criteria are summarized in Table 3.

## 5. Future Directions and Challenges

The advent of targeted KIT inhibition has transformed the treatment landscape for patients with AdvSM. Patients are achieving complete responses and molecular remissions, which have not been seen previously. This has translated into improved quality of life and overall survival for patients with this rare life-limiting disease. However, there is still work to do as up to 20% of patients experience disease progression, usually in the AHN component [47]. The identification of further therapeutic targets is important to address this unmet need.

The immune checkpoint proteins programmed death-1 (PD-1) and its ligand (PD-L1) form the PD-1/PDL-1 pathway, which protects tissues from immune-mediated damage and leads to the inhibition of cytotoxic T cells in the tumour micro-environment, inhibiting tumour destruction [48]. PD-L1 is expressed in mastocytosis, as demonstrated by immunohistochemistry and multiplex fluorescence staining [49]. In 77% of SM cases, neoplastic MCs display membrane staining with PD-L1, with rates of 100%, 100%, 75% and 50% for MCL, ASM, SM-AMH and ASM, respectively [50]. Therefore, it represents a potential novel therapeutic target for AdvSM and supports the research of therapeutic agents with activity against PD-L1 in patients with AdvSM. PD-1 inhibitors (e.g., Pembrolizumab and Nivolumab) are already used to treat other haematological malignancies [51].

Characterising the molecular profiles or signatures to guide changes in therapeutic strategies in patients with AdvSM with an AHN will be extremely informative. Utilising the mutation VAF and impact of therapies will be useful in guiding treatment changes to target the appropriate clone, be it sequential treatment options for SM/AHN or ideally using a combination of doublet or triplet regimes. This would be particularly useful for high-risk patients to optimise responses and the time to response, as well as potentially increasing the possibility of patients moving to an alloHSCT option if available. The future use of single-cell genomics (mast cell or AHN DNA/RNA sequencing) will add to the granularity and understanding of the changing bone marrow micro-environment in these complex patients.

## 6. Conclusions

The advent of the TKI era has revolutionized the treatment landscape for patients with advanced systemic mastocytosis and, more recently, those with symptomatic indolent systemic mastocytosis. Avapritinib significantly improves OS and quality of life in AdvSM patients and can achieve deep molecular remission of *KIT* D618V mutations. Preliminary trail data suggests that Bezuclastinib exhibits signs of disease-modifying potential. The use of NGS allows for early mapping of an individual’s mutational profile, thereby enabling the application of prognostic scoring systems and guiding treatment decisions. Effectively treating the subvariant SM-AHN, with its associated multimutated landscape beyond *KIT* D816V, remains a significant challenge. The role of combining KIT inhibitors with AHN-directed therapy in this cohort is a question currently being addressed in clinical trials and warrants ongoing focus. However, it is important to highlight that not all patients with SM-AHN require upfront targeted treatment beyond supportive care. Instead, an expectant “watch and wait” management approach may be appropriate for a small cohort. For example, a patient with ISM-AHN (no C-findings) with an AHN characterised by a favourable risk profile, such as low-risk ET or CMML without excess BM blasts and no high-risk mutations, can simply be monitored for symptoms and signs of progression in either component. Large international collaborative efforts are required to address the remaining questions regarding the therapeutic management of advanced systemic mastocytosis.

## Figures and Tables

**Figure 1 diagnostics-14-00080-f001:**
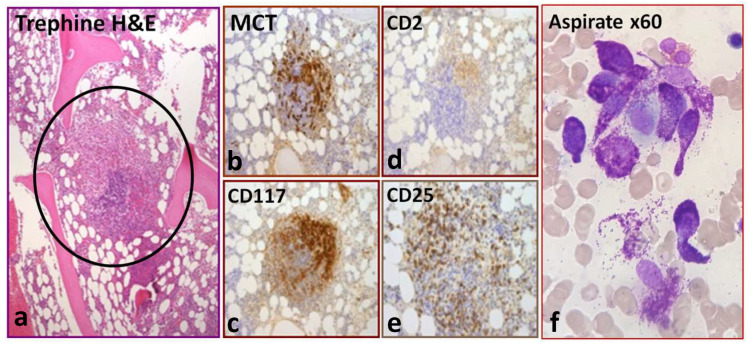
Typical bone marrow histology, immunohistochemistry and aspirate morphology. (**a**) Bone marrow trephine H&E section showing an aggregate of greater than 15 mast cells (circled) surrounded by lymphocytes and eosinophils (major criterion). The middle panels (**b**–**e**) show the presence of clonality by positive immunohistochemistry staining with mast cell tryptase (MCT), CD2, CD117 and CD25 (minor criteria). (**f**) Aspirate with the classical spindle-shaped morphology of abnormal mast cells.

**Figure 2 diagnostics-14-00080-f002:**
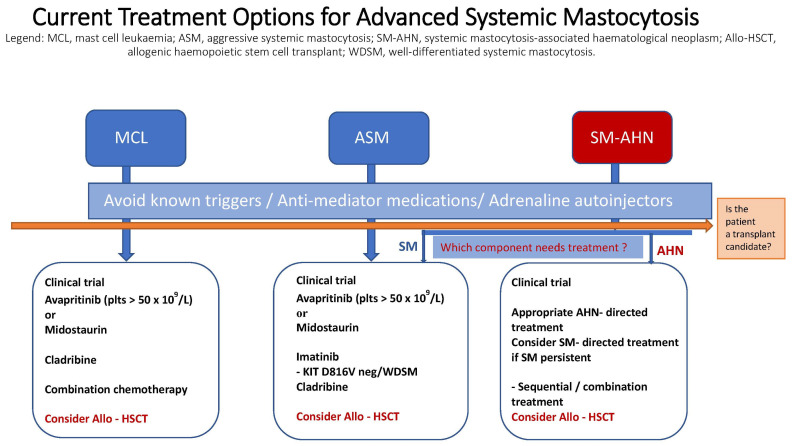
Current treatment options for advanced systemic mastocytosis.

**Table 1 diagnostics-14-00080-t001:** The WHO and ICC diagnostic criteria for SM.

World Health Organisation (WHO) Requires at Least 1 Major and 1 Minor or 3 Minor Criteria
Major criterion Multifocal dense infiltrates of MCs (≥15 MCs in aggregates) in BM biopsies and/or in sections of other extracutaneous organ(s)
Minor criteria >25% of all MCs are atypical cells (type I or type II) on BM smears or are spindle-shaped in MC infiltrates detected on sections of visceral organ KIT point mutation at codon 816 in the BM or another extracutaneous organ MCs in BM or blood or another extracutaneous organ exhibit CD2 and/or CD25 Baseline serum tryptase level >20 ng/mL In case of an unrelated myeloid neoplasm, item d is not valid as an SM criterion
International Consensus Criteria (ICC) The major criterion alone is sufficient for diagnosis. In the absence of the major criterion, at least 3 of the 4 minor criteria are required
Major criterion Multifocal dense infiltrates of tryptase- and/or CD117-positive mast cells (≥15 mast cells in aggregates) detected in sections of bone marrow and/or other extracutaneous organ(s)
Minor criteria In BM biopsy or in section of other extracutaneous organs >25% of mast cells are spindle shaped or have an atypical immature morphology MCs in BM, PB or other extracutaneous organs express CD25, CD2 and/or CD30, in addition to MC markers *KIT* D816V mutation or other activating *KIT* mutation detected in BM, PB or other extracutaneous organs Persistently elevated serum tryptase level >20 ng/mL In cases of SM-AMN, an elevated tryptase does not count as a SM minor criterion.

**Table 2 diagnostics-14-00080-t002:** Prognostic scoring systems for AdvSM. MARS: Mutation-Adjusted Risk Score; GPSM: Global Prognostic Score for Mastocytosis; PFS: progression-free survival; OS: overall survival; IPSM: International Prognostic Scoring System for Mastocytosis; MAPS: Mayo Alliance Prognostic System.

	Prognostic Scoring System			
	MARS	GPSM-PFS ^a^ GPSM-OS ^b^	IPSM	MAPS
Age ≥ 60 years	x		x	x
Haemoglobin g/L				
≤100	x			
≤110		b	x	
Platelet count × 10^9^/L				
≤100	x	a	x	
≤150				x
Leucocyte count × 10^9^/L				
≥16			x	
Increased serum				
Alkaline phosphatase		b		x
Serum β2m ≥ 2.5 μg/mL		a	x	
Serum tryptase ≥ 125 ng/mL		a	x	
Additional somatic mutations	SRSF2	SRSF2 ^a^		
	ASXL1	ASXL1 ^a^		
	RUNX1	RUNX1 ^a^		
		DNMT3A ^a^		

**Table 3 diagnostics-14-00080-t003:** The modified IWG-MRT-ECNM Response Criteria.

**Complete remission (CR)**
All four criteria of a duration ≥ 12 weeks: - No compact neoplastic MC aggregates in BM or extracutaenous organ - Serum tryptase level < 20 ng/mL - Peripheral blood count remission [ANC ≥ 1 × 10^9^/L, Hb level ≥ 11 g/dL, PLT ≥ 100 × 10^9^/L] - Complete resolution of palpable hepatosplenomegaly and all biopsy-proven or suspected SM-related organ damage
**CR with partial recovery of peripheral blood counts (CRh)**
All CR criteria met ≥12 week duration but with residual cytopenias. The following minimum recovery is required: ANC > 0.5 × 10^9^/L, Hb level > 8.0 g/dL, PLT > 50 × 10^9^/L
**Partial remission (PR)**
All three criteria of a duration ≥12 weeks: - Reduction by ≥50% in neoplastic MCs in the marrow and/or or other extracutaneous organs showing SM-related organ damage - Reduction of serum tryptase level by ≥50% - Resolution of one or more biopsy-proven or suspected SM-related organ damage
**Clinical improvement (CI)**
Response duration must be ≥12 weeks Requires more than one of the nonhaematological and/or haematologic response criteria to be fulfilled in the absence of CR, CRh, PR or PD
**Stable disease (SD)**
Not meeting criteria for CR, CRh, PR, CI or PD
**Progressive disease (PD)**
Requires worsening of at least one specified criteria; duration must be ≥4 weeks. The list of criteria are well-defined [45,46]
**Loss of response (LOR)**
Loss of a documented CR, CRh, PR, or CI or downgrading of CR/CRh for ≥8 weeks

## Data Availability

No new data were created or analyzed in this study. Data sharing is not applicable to this article.

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
