# Peer review of "Recent Advances in the Therapeutic Management of Advanced Systemic Mastocytosis"

_diagnostics, 2023, doi:10.3390/diagnostics14010080_

Round 1

Reviewer 1 Report

Comments and Suggestions for Authors

This is a review of “Recent Advances in the Therapeutic Management of Advanced Systemic Mastocytosis” by S. Veitch and D.H. Radia.  This is a narrative review that focuses on the clinical management of systemic mastocytosis (SM), although the authors also address the current diagnostic criteria and prognostic scoring systems as well.  As a laboratory medicine physician, I am not qualified to critique the portions of the manuscript that discuss therapeutic aspects of SM, so I will confine my remarks to the parts of the manuscript that are pertinent to my area of expertise.

1)     The authors do a good job highlighting the changes to the pathologic diagnostic criteria in the new WHO and the ICC systems (lines 45-51).  However, it may be worthwhile to summarize the diagnostic criteria for these two systems in a table.  Could you also add references to the WHO and ICC classifications?

2)     I am not sure that I would attribute the mismatch between the aspirate and core biopsy mast cell percentages solely to suboptimal bone marrow aspirate technique (line 52).  To be sure, suboptimal bone marrow aspirates are a significant problem, especially in tertiary care centers and other training hospitals.  However, in the case of SM, features of the disease itself such as 1) focality of the mast cell lesions and 2) localization of the mast cell infiltrates within areas of fibrosis, may be even greater contributors to the disparity between aspirate and core biopsy findings.

3)     Since the authors discuss the pathologic diagnosis of SM, it may be useful to include an image of the morphologic findings in a typical case, including (if possible) images demonstrating abnormal mast cell morphology and aberrant expression of CD2, CD25, or other antigens.

4)     For the discussion of prognostic scoring systems (lines 83-88) it may be useful to include a table that compares the systems.

5)     What is the recommended follow-up for patients with SM?  It may be helpful to include recommendations including 1) intervals for follow-up studies 2) when/ whether a bone marrow biopsy is indicated in all cases 3) the role of flow cytometry/ molecular testing for measurable residual disease.

Reviewer 2 Report

Comments and Suggestions for Authors

The review articel comprehensively describes the latest developments in the treatement of systemic mastocytosis with a focus in the new KIT D816V targeting drugs. The literature is balanced reviewed and the article is well written.

I only have some minor comments:

1) The introduction section summarizes the updates of diagnostic criteria in WHO HAEM-5. The update on basal serum tryptase levels with correction for the tryptase genotype should be briefly mentioned as well.

2) Italicization of gene names is currently only used for KIT. It should also be used for the other gene names, e.g. TET2, SRSF2, ASXL1, RUNX1, ...

3) The abbriviation PK might not be generally known and should be spelled out when used first in the manuscript.

4) A brief summary of the updated treatment response criteria for advanced and indolent SM in the introduction section could be helpful for the reader.

Round 2

Reviewer 1 Report

Comments and Suggestions for Authors

Thank you for your revision.